# Biostimulants in Corn Cultivation as a Means to Alleviate the Impacts of Irregular Water Regimes Induced by Climate Change

**DOI:** 10.3390/plants12132569

**Published:** 2023-07-07

**Authors:** Gabriel Luiz Piati, Sebastião Ferreira de Lima, Renato Lustosa Sobrinho, Osvaldir Feliciano dos Santos, Eduardo Pradi Vendruscolo, Janaina Jacinto de Oliveira, Tassila Aparecida do Nascimento de Araújo, Khairiah Mubarak Alwutayd, Taciane Finatto, Hamada AbdElgawad

**Affiliations:** 1Department of Agronomy, Federal University of Mato Grosso do Sul (UFMS), Chapadão do Sul 79560-000, Mato Grosso do Sul, Braziljanaina.jacinto@ufms.br (J.J.d.O.); 2Department of Agronomy, Federal University of Technology—Paraná (UTFPR), Pato Branco 85503-390, Paraná, Braziltfinatto@utfpr.edu.br (T.F.); 3Department of Agronomy, State University of Mato Grosso do Sul (UEMS), Cassilândia 79540-000, Mato Grosso do Sul, Brazil; 4Department of Biology, College of Science, Princess Nourah bint Abdulrahman University, Riyadh 11671, Saudi Arabia; Kmalwateed@pnu.edu.sa; 5Integrated Molecular Plant Physiology Research, Department of Biology, University of Antwerp, 2000 Antwerp, Belgium; hamada.abdelgawad@uantwerpen.be; 6Botany and Microbiology Department, Faculty of Science, Beni-Suef University, Beni-Suef 62521, Egypt

**Keywords:** climate change, irregular water regimes, *Zea mays*, phytohormones, water deficit tolerance

## Abstract

Climate change alters regular weather seasonality. Corn is one of the main crops affected by irregular water regimes. Due to complications in decision-making processes related to climate change, it is estimated that planting corn outside the optimal window results in around USD 340 million in losses per year in the United States’ Corn Belt. In turn, exogenous plant growth regulators have been gaining prominence due to their potential to positively influence the morphology and physiology of plants under stress. This study was based on the hypothesis that the use of plant growth regulators can assist in mitigating the adverse effects of climate change on corn plants sown both inside and outside the recommended planting period. In this context, the effects of biostimulant application on gas exchange in corn plants sown within and outside the recommended period were evaluated. The experiment was carried out in randomized blocks in a 4 × 5 × 2 factorial scheme with four repetitions. These were four sowing times, the application of the biostimulants via seeds in five doses, and foliar applications (presence and absence). The biostimulant doses were 0.00, 6.25, 12.50, 18.75, and 25 mL kg^−1^. The foliar application used a dose of 500 mL ha^−1^. Only in the period (2017/2) higher doses of biostimulants indicated a decrease in the water use efficiency of plants, suggesting the need to evaluate this variable carefully. In this regard, future studies may investigate the ideal doses and application timings of biostimulants for different edaphoclimatic conditions. In general, the combined use of biostimulants on seeds and as a foliar treatment boosted physiological activity and stimulated photosynthetic processes in corn plants. Based on these data, plant regulators can be a useful tool to mitigate the adverse effects of climate change on corn plants sown inside and outside the planting period.

## 1. Introduction

Climate change is increasing the likelihood of extreme weather events, presenting significant challenges for the planning of agricultural planting, which heavily depends on regular climatic seasonality [1]. As a result, a decrease in the yield of various crops has been observed. In this context, rainfall regimes can be considered one of the main factors directly influencing the cultivation of commercial species outdoors, becoming an obstacle when there is an excess or deficit of rainfall [2].

Corn, mainly cultivated outdoors, is one of the main crops affected by irregular water regimes, complicating decision making about planting and cultural management [3]. In the case of corn, a lack of soil moisture can cause various morphophysiological changes, such as an increase in oxidative activity, a decrease in photosynthetic rates, and a reduction in productivity [4,5]. Therefore, climatic irregularity and the biotic stress associated with it represent a potential threat to global food security [6].

Although plants have natural morphological and physiological mechanisms that help to minimize the negative effects of abiotic stress, research in areas such as genetics, nutrition, and plant physiology have sought the practical use of products that can mitigate the effects of this type of stress in plants [7,8]. Thus, the use of exogenous plant regulators has gained prominence due to their potential ability to influence the morphology and physiology of plants under stress, aiming to improve their growth potential and productivity, even under normal or abiotic stress conditions [9,10].

In this sense, a study conducted by Mitch et al. [11] assessed how climate change complicates the decision-making process for choosing the ideal planting time for corn cultivars in the central region of the United States’ Corn Belt; these authors found losses of around USD 340 million per year in the region due to corn planting outside of the optimum window. Another study conducted by Malik et al. [12] explores the possibility of using biostimulants to address some of the challenges of climate change, especially highlighting their potential to reduce the effects of water deficit.

However, the study by Malik et al. [12] also revealed that farmers’ understanding of the functions and potential applications of biostimulants is still limited, constituting a barrier to the establishment of more productive and sustainable agricultural practices in the face of challenges posed by climate change. Furthermore, the biostimulants available on the market are diverse and can also vary in relation to application methodologies. In this context, the study by Barbosa et al. [13] reinforces the need for the development of more research evaluating how corn plants respond to different biostimulants in different field situations involving different application methodologies.

This study evaluated the hypothesis that the use of growth regulators can be a useful tool to mitigate the negative impacts of climate change on the physiological performance of corn plants sown both inside and outside of the optimum planting period. Although there have been many laboratory studies on the stress-mitigating effects of plant growth regulators, this effect is rarely studied under field conditions [14,15]. Compared to laboratory studies, field studies provide a more holistic perspective by considering the complexities and variability of real-world conditions. Thus, field studies are essential for understanding the complex interactions between growth regulator hormones and plant stress responses. Our study will also provide valuable insights into the practical applications of a mixture of growth-promoting hormones (kinetin, gibberellin, indolebutyric acid) for stress tolerance of corn in agriculture, while accounting for the genetic, environmental, and ecological complexities that exist in natural systems.

## 2. Results

### 2.1. Complete Analysis of Variance and Gas Exchange of Corn as a Function of Foliar Application of Biostimulant

For corn sown in February 2016 (2016/1), until the evaluation of gas exchange in plants (ATG), 438 mm of rainfall was recorded, and only one day in which the crop remained at the permanent wilting point (PM). For corn sown in March of the same year (2016/2), the period from PM of the crop to the date of ATG was eighteen days, with precipitation of 152 mm until the phenological stage V8. In the year 2017, there was an accumulation of precipitation until the ATG of 463 and 301 mm, for corn sown in February (2017/1) and March (2017/2), respectively, with a PM of seven days in 2017/1 and ten days in 2017/2 (Figure 1).

There was a significant interaction between all factors for the variables of net photosynthesis rate, stomatal conductance, instantaneous carboxylation efficiency, and leaf area (Table 1). The sub-stomatal CO_2_ concentration and instantaneous water use efficiency showed a significant interaction for sowing time × biostimulant via foliar and sowing time × biostimulant via seed. Transpiration showed no significant interaction only for the following combination: sowing time × foliar biostimulant (Table 1).

In sowing 2016/2, despite the corn crop having accumulated the highest number of days of water deficit of all seasons until the moment of the evaluations (18 days), the results of the foliar application show that the biostimulant favored an increase in the net photosynthesis rate (Table 1), also reflecting the instantaneous efficiency of both water use and carboxylation, as these variables are dependent on the effectiveness in the assimilation of CO_2_ by plants. They also favor the vegetative development of the culture represented by the variable LA (Table 2). Such results were correlated since the plants were not subjected to water deficit at the time of data collection (Figure 1).

Because the sowing times presented different water availability at the time of the evaluations (Figure 1), unlike the 2016/2 season, the foliar application of a biostimulant in the later cultivation of corn in 2017 (2017/2) caused a significant decrease in some physiological aspects of plants, including net photosynthesis, stomatal conductance, and the instantaneous efficiency of water use and carboxylation (Table 2). This demonstrated the hormonal imbalance caused by the negative physiological effects on plants [16] that, in this study, occurred at a time when the corn crop was subjected to conditions of intense water restriction, associated with the fact that, in the 2017/2 season, the plants were at the wilting point in the last two days before the evaluation of their gas exchange.

Although the instantaneous efficiency of carboxylation was lower in plants that received the foliar application of the biostimulant in the 2017/1 season, they could internally retain a greater amount of CO_2_, a fact that can be explained by the higher stomatal conductance (Table 2) because it is indirectly associated with the degree of stomata opening, and consequently the higher level of carbon dioxide.

The highest value of the variable *Ci* obtained through the presence of the biostimulant in corn plants in the 2017/1 season is considered a positive factor, since it is known that CO_2_ is a substrate for photosynthesis, which can interfere with this process. Even so, despite the numerical difference, statistically variable A was the same regardless of the foliar application of the biostimulant in 2017/1 (Table 2).

As in 2017/1, a higher stomatal conductance was also observed in corn with the foliar application of a biostimulant in 2016/1 (Table 2), but this difference did not significantly affect any other physiological aspects of the plants, unlike what was observed in the 2017/1 sowing. The evaluation carried out in the first season of 2016 was under conditions of greater water availability for the plants, concerning the other seasons (Figure 1), which suggests that, to some extent, the greatest potential effect of the foliar application of a biostimulant occurs in conditions that are more restrictive to vegetative development.

The relevance of the water availability factor for the crop between the different sowing times in most of the tested variables was clear, observing higher values for net photosynthesis, sub-stomatal CO_2_ concentration, instantaneous carboxylation efficiency, and leaf area in 2016/1, in which the last mentioned variable had no significant difference with 2017/1. Except for stomatal conductance and instantaneous efficiency of water use, it was noticed that the variables followed the order of rainfall volume in each season, as the best physiological results were presented at the most favorable season and year for corn 2016/1 followed by 2017/1, 2016/2 and 2017/2 (Table 2).

### 2.2. Corn Gas Exchange as a Function of Biostimulant Application Doses via Seed in Different Periods

In general, the doses of the biostimulant in the seeds caused improvements in the physiological aspects of the plants, as in variables A and *Ci* (Figure 2A,D). The plants sown in the 2016/1 season were superior to the others in terms of CO_2_ assimilation, probably, as previously emphasized, because they are in a more favorable environment for their development, where a dose of 13.38 mL of biostimulant per kg of seeds provided an increase of 8.73%, reaching 44.96 µmol of CO_2_ m^−2^ s^−1^. The plants from the 2017/1 and 2017/2 seasons showed maximum rates of *A* with doses of 13.78 and 10.04 mL of the biostimulant per kg of seeds, with increases of 19.12 and 9.21% in photosynthesis, reaching 39.05 and 30.85 µmol of CO_2_ m^−2^ s^−1^, respectively (Figure 2A).

Different behavior for variable *A* was observed in 2016/2, in which a higher dose of biostimulant, when applied to seeds, resulted in greater assimilation of CO_2_ by plants, with approximately a 36.9% increase in net photosynthesis (35.88 µmol of CO_2_ m^−2^ s^−1^) with a dose of 25 mL kg^−1^ of seeds. At that time, the application of biostimulant had greater effects concerning the others, but the highest photosynthetic activities were only evidenced in high doses of the biostimulant. Low doses of the biostimulant did not allow the values of variable *A* to be higher than those obtained, especially at times when sowing was carried out in February, probably due to the water deficit resulting from the eighteen days in which the plants remained at the wilting point in the 2016/2 season.

The stomatal conductance of plants increased with increasing doses of the biostimulant in 2016/2, with 0.58 mol of H_2_O m^−2^ s^−1^ through the application of the highest dose of biostimulant in corn seeds, an increase of approximately 65%(Figure 2C). This may be related to the increased tolerance of plants to water deficit caused by the application of the biostimulant, as the greater the stomatal opening, the greater the influx of CO_2_ into the plants for the photosynthesis process.

The physiological ability to adapt the corn crop to different environmental conditions was found in the transpiration results at different times (Figure 2B), as lower values in this variable were obtained in times with longer periods of plant water deficit (2017/2 and 2016 /2). On the other hand, the increase in *gs* and CO_2_ assimilation, some doses of the biostimulant led to a significant increase in plant transpiration.

The increase in variable *E* was linear in both seasons of 2016 (Figure 2B), where the highest dose of the biostimulant provided increases of 28.89 and 35.88% for 2016/1 and 2016/2, respectively, with values approximately twice as high in the first sowing of that year. In 2017, doses of 12.5 and 14.42 mL kg^−1^ were responsible for the highest *E* values found, with increases of 17.79 and 14.73% in 2017/1 and 2017/2, respectively.

For *gs* in 2017, the doses of 13.14 and 15.33 mL kg^−1^ were responsible for the highest values in stomatal conductance, which can be related to the opening of stomata in 2017/1 and 2017/2, respectively.

On the other hand, the expressive values found in the variable *gs* in 2016/2 did not reflect a higher transpiration rate, and for the first season of 2016, there was a behavior similar to 2017, with an increase of 21.66% in stomatal conductance via the application of 15.2 mL kg^−1^ (Figure 2C).

Levels above 250 µmol CO_2_ mol^−1^ air, in the sub-stomatal CO_2_ concentration were observed with the application of 22.96 mL of biostimulant per kg of seeds when sowing was carried out in 2016/1. Despite the decrease of 6.13% with the highest dose of biostimulant, the values of the variable *Ci* in 2017/1 were still 14.28 and 27.2% higher than in 2016/2 and 2017/2, respectively. In the two seasons that corresponded to the lowest *Ci* values, there was an average increase of 4.45% in *Ci* with the highest dose in 2016/2, corresponding to 196.27 µmol CO_2_ mol^−1^ air and 4.02% with 12, 24 mL of biostimulant per kg of seeds, obtaining 175.97 µmol CO_2_ mol^−1^ air (Figure 2D).

The highest dose of the biostimulant in the seed treatment caused decreases of 28.57% and 5.36% in the variable *EICI* in 2016/1 and 2017/2, respectively (Figure 2F).

The increase in *Ci* resulting from the application of the biostimulant was higher than the increase in the net photosynthesis rate. This explains the decrease in the instantaneous efficiency of carboxylation, as it is a variable arising from the relationship between variables *A* and *Ci*.

In the other seasons, there was an increase of more than 20% in the *EICI* with doses of 25 and 15.13 mL kg^−1^ of the biostimulant for the 2016/2 and 2017/1 seasons, respectively.

As previously described, the use of the biostimulant caused an increase in the variable *gs*, which has a strong relationship with transpiration (2017/1). The decrease in *IWUE* in 2016/1 was 20.62% with the maximum dose of biostimulant (25 mL kg^−1^), with an efficiency of 3.31 µmol m^−2^ s^−1^/mmol of H_2_O m^−2^ s^−1^. For 2016/2 and 2017/2, the lowest efficiency occurred with doses of 12.09 and 21.28 mL of the biostimulant, which resulted in 4.46 and 5.12 µmol m^−2^ s^−1^/mmol of H_2_O m^−2^ s^−1^, respectively (Figure 2E).

The highest value for leaf area with the use of the biostimulant was found in the later sowing of 2016 (2016/2), with an average growth of 10% with the application of 15.59 mL kg^−1^ in 2016/1 and 13, 72 mL kg^−1^ of biostimulant in seeds in 2017/1. In 2016/2, the increase achieved in corn leaf area was approximately 16% with a dose of 17.85 mL kg^−1^ of the product, partially confirming one of the hypotheses addressed in the development of the present study, in which the response potential in the application of biostimulant may be higher under stressful conditions for plants, with reservations, because when observing the time when there was the greatest intensity of water deficit for plants (2017/2), the efficiency of the biostimulant in the variable LA was about 50% lower than in 2016/2.

### 2.3. Corn Gas Exchange as a Function of Biostimulant Application Doses via Seed Combined with Foliar Application

Gas exchange was also affected by the presence or absence of foliar application of the biostimulant at the V4 stage of corn crop (Figure 3A), demonstrating the physiological benefit provided by the interaction between seed and foliar application modes.

With the use of biostimulant via foliar, the dose of 15.52 mL kg^−1^ applied via seeds was responsible for higher values of variable *A* (36.99 mmol of CO_2_ m^−2^ s^−1^) (Figure 3A). Regardless of the foliar application of the biostimulant, the doses in the seeds provided an increase of more than 10% in the net photosynthesis rate. *E* resulted in maximum water loss, 9.03 mmol of H_2_O, with a dose of 15.54 mL of biostimulant applied to the seed, in the absence of foliar application. When the biostimulant was applied via the leaves, the maximum transpiration was 9.35 mmol of H_2_O, with a dose of 20.12 mL kg^−1^ of biostimulant in the seeds (Figure 3B).

Without the application of the biostimulant via foliar, there was a significant increase in stomatal conductance up to the dose of 13.3 mL kg^−1^, with a maximum value of 0.51 mol of H_2_O m^−2^ s^−1^, 17.18% higher than control. However, the application of biostimulant via foliar stimulated greater responses in *gs* values, reaching 0.56 mol of H_2_O m^−2^ s^−1^ with a dose of 20.25 mL kg^−1^, representing an increase of 29.45% in the values of this variable (Figure 3C).

By the application of biostimulant positively influencing the physiological activity of corn plants, it was expected to find greater efficiency in the use of substrate CO_2_ in the photosynthetic process, however, the use of foliar biostimulant enhanced the results. The variable *EICI* had an increase in its values up to the dose of 15.83 mL kg^−1^ of biostimulant applied to the seeds, reaching an almost 10% increase in the values of this variable when the foliar application of this product was utilized. For treatments in which the biostimulant was used only in seeds, a dose of 22.5 mL kg^−1^ increased the values of this variable by only 5.91% (Figure 3D).

The effect of the application of biostimulant via foliar on the vegetative development of the corn crop is evidenced in the results for the leaf area, showing an average superiority of 13.33% in this variable concerning the treatments that did not receive the application of the product. The application of the biostimulant also potentiated the effect of the same product when used in seed treatment, increasing plant leaf growth by 12.23%, with a dose of 14.63 mL kg^−1^, which resulted in 4743.09 cm^−2^ plant^−1^, while in the absence of foliar application, the maximum increase achieved in the LA variable was 4103.29 cm^−2^ plant^−1^, 7.19% higher than the control, obtained via the application of 14.43 mL kg^−1^ (Figure 3E).

## 3. Material and Methods

### 3.1. Experimental Area, Soil and Weather Conditions

The experiment was conducted during the second crop seasons of 2016 and 2017 in the experimental area of the Federal University of Mato Grosso do Sul, Chapadão do Sul Campus, with a latitude of 18°47′39″ S, longitude of 52°37′22″ W, and altitude of 820 m. In each year of cultivation, an experiment was conducted in February and another in March, totaling four experiments.

According to Cunha et al. [17], the climate of the region is classified as humid and tropical, with dry winters and rainy summers. The water balance (Figure 1) was performed using meteorological data, where Crop Evapotranspiration (ETc) was measured as the product of the Reference Evapotranspiration (ETo) and the Crop Coefficient (Kc). The ETo estimates were obtained using the Penman–Monteith–FAO method, according to Allen et al. [18], using data from an automatic weather station (Code A730) of the Instituto Nacional de Meteorologia (INMET).

The soil in the experimental area was classified as a dystrophic Red Latosol, with a clayey texture [19], a density of 1.2108 g dm^−3^ and a water content equivalent to the field capacity and permanent wilting point of the plant: 0.2652 and 0.1858 dm^3^ per dm^−3^, respectively.

In 2016, during the chemical analysis of the soil layer from 0.00 to 0.20 cm, the following results were found: 9.0 mg dm^−3^ of P (Melich); 33.5 g dm^−3^ of organic matter (O.M) with pH 4.9 (CaCl_2_); K^+^, Ca^2+^, Mg^2+^ and H + Al = 0.07, 2.40, 0.9, and 2.9 cmol_c_ dm^−3^, respectively; and 53.7% base saturation. In 2017, in the same soil layer, the following results were found: 8.8 mg dm^−3^ of P (Melich); 28.0 g dm^−3^ of O.M; pH 4.9 (CaCl_2_); K^+^, Ca^2+^, Mg^2+^, and H + Al = 0.24, 2.10, 0.90, and 3.8 cmol_c_ dm^−3^, respectively; and a base saturation of 46.37%.

### 3.2. Experimental Design and Treatments

The experimental design was used in randomized blocks in a 4 × 5 × 2 factorial scheme with four replications, corresponding to sowing corn four times, the application of a biostimulant via seeds in five doses, and the application of a biostimulant via foliar (presence and absence). The sowing seasons were divided into two agricultural years: 2016 and 2017. For 2016, sowing took place on 5 February (2016/1), and 8 March (2016/2), and in 2017 on 15 March, February (2017/1), and 9 March (2017/2). The biostimulant doses applied to seeds were 0.00 (Control), 6.25 (Dose 1), 12.50 (Dose 2), 18.75 (Dose 3), and 25.00 (Dose 4) mL kg^−1^. The application of a biostimulant via foliar was carried out at the V4 stage (four fully expanded leaves) at a dose of 500 mL ha^−1^. The experimental plots were formed by five rows of corn, spaced 0.45 m apart and 5 m long, resulting in a total area of 11.25 m^2^ and a useful area of 4.05 m^2^.

### 3.3. Experimental Conduction and Evaluations

The biostimulant used was a solution comprising three phytoregulators, containing 0.009% kinetin (cytokinin), 0.005% gibberellic acid (gibberellin), and 0.005% indolebutyric acid (auxin). The application in the seeds was carried out two hours before the sowing of the culture. The product was applied directly to the seeds that were packed in plastic bags and shaken for two minutes to homogenize them. The application via foliar was carried out with respect to the ideal environmental conditions for the maximum absorption of the product by plants (temperature range from 20 to 25 °C, 70% relative humidity, and wind speed below 10 km h^−1^) and with a flow of 150 L ha^−1^, using an electric constant pressure sprayer.

One week before sowing corn in the two years of cultivation, the area was desiccated using the herbicide, Diquate (0.5 kg a.i. ha^−1^), and mineral oil (0.321 kg a.i. ha^−1^). The areas were cultivated before sowing corn with soybeans, corresponding to the 2015/16 and 2016/17 harvests. The sowing system of no-tillage, without irrigation, was used, which involved placing 3 seeds per meter, resulting in a density of 66.6 thousand seeds per ha^−1^. The seed used was the single hybrid AG 8061 VT PRO YieldGard^®^, being treated before sowing with Pyraclostrobin (0.005 kg a.i. 100 kg^−1^), Methyl Thiophanate (0.045 kg a.i. 100 kg^−1^) and Fipronil (0.05 kg a.i. 100 kg^−1^). The management of corn fertilization at sowing and in coverage adhered to the needs identified via a chemical analysis of the soil, according to Sousa and Lobato [20]. The sowing of crops concerning the management of weeds, pests and diseases followed EMBRAPA [21].

The evaluation of gas exchange was carried out at the V8 phenological stage (eight fully expanded leaves), from 9 am to 11 am, and the middle third of the last fully expanded leaf was measured using infrared IRGA (Licor Li 6400 XT model, LI-Cor) with an airflow of 500 μmol s^−1^ and coupled light source of 1000 μmol m^−2^ s^−1^. On this occasion, the net photosynthesis rate (*A*) (μmol of CO_2_ m^−2^ s^−1^), transpiration (*E*) (mmol of H_2_O m^−2^ s^−1^), stomatal conductance (*gs*) (mol of H_2_O m^−2^ s^−1^) and sub-stomatal CO_2_ concentration (*Ci*) (μmol CO_2_ mol^−1^ air) were calculated. The instantaneous water use efficiency (*IWUE = A/E)*—(μmol of CO_2_ m^−2^ s^−1^)/(mmol of H_2_O m^−2^ s^−1^)—was calculated with regard to net photosynthesis with transpiration and instantaneous carboxylation efficiency (*EIC* = *A*/*Ci*)—(μmol of CO_2_ m^−2^ s^−1^)/(μmol CO_2_ mol^−1^ air)—from the relationship between net photosynthesis and sub-stomatal CO_2_ concentration. At the same phenological stage, the average leaf area per plant was measured using a portable leaf area meter, choosing three plants per plot to calculate the average of the treatments.

### 3.4. Statistical Analysis

The assumptions of normality distribution and homogeneity of variances were verified for the data. The data were subjected to an analysis of variance with the means of the qualitative factors, compared using the Tukey test. The means of the quantitative factors were evaluated using regression analysis, both at 0.05 probability.

## 4. Discussion

Studies report that the application of a biostimulant via foliar causes an increase in the absorption of water and nutrients by plants [22,23]. It is known that one of the functions of the cytokinin hormone is that it stimulates the mobilization of nutrients according to crop demand in the same way that auxin is linked to increased nutrient demand by stimulating vegetative growth [16,24]. Therefore, due to the hormonal composition present in the hormonal biostimulant, its foliar application may provide greater availability of nutrients and better use of water by the crop, positively influencing the increase in photosynthetic rates in the 2016/2 season. Future studies may investigate the relationship between the application of biostimulants and the uptake of various nutrients by plants.

The use of this biostimulant or the increase in its dose harmed or did not influence crops such as cotton [25] and soybean [26], indicating that the product may have adverse effects, depending on the crop or the cultivation environment. However, in corn, despite the decrease in the physiological activity of the plants via the foliar application of biostimulant in the 2017/2 season, there was a significant increase of 7 to 8% in leaf area (Table 2), which may compensate for the negative effects shown in the 2017/2 seasons, gas exchange in 2017/2, or increase the efficiency in the production of photoassimilates by plants in the rest of the seasons, in which the presence of the biostimulant was more favorable for the physiological aspects in corn.

The highest *IWUE* of corn plants in the 2017/2 seasons, followed by the 2016/2 (Table 2), proved the efficiency in water use in periods of greater water deficit compared to the other seasons (Figure 1). Similar behavior has already been reported in other studies [27,28]. This effect may also be related to the greater development of the root system, obtained via the application of the biostimulant, given the presence of auxins, the main hormone involved in root growth [16,29,30]. Auxin has already been detected in corn crops subjected to salinity stress, in which the biostimulant provided an increase of more than 70% on the dry mass of roots when the plants were treated via seeds with a dose of 13 mL kg^−1^ [31]. Future studies may investigate the relationship between the application of biostimulants and morphological changes in plants, such as root size and growth.

Severe and prolonged water deficit can directly impact photosynthesis, inducing plants to respond to water stress via several adaptations, such as the inhibitory action in the biochemical phase of the photosynthesis process [7,32,33], as well as changes in endogenous levels of plant hormones through increased abscisic acid levels and decreased auxin and gibberellin levels in corn plants [34]. This correlates with the results obtained in this study and justifies the importance of hormonal balance in increasing plant tolerance under water stress conditions through the exogenous application of these hormones [7,8].

Similar to what was observed for *A*, the effect of water deficit on stomatal conductance was observed with greater emphasis on the lowest doses of biostimulants in seeds for the 2016/2 season, as these were the lowest values found. Plants exposed to water deficit use the stomatal closure mechanism to reduce the loss of water through transpiration to the atmosphere [8,35], this process is dependent on hormonal signals, with an increase in abscisic acid and an abrupt reduction in gibberellin levels in plants [34]. Since the used biostimulant contains gibberellic acid, this may have influenced the stomatal opening process, thus helping to increase the tolerance of the stomata of corn to water stress [29,30].

In situations of optimal water availability, plants generally have high transpiration rates. However, as water in the soil becomes scarce, there will be a reduction in the transpiration rate of plants to minimize losses, and thus conserve the available water in the soil [36,37], a fact that may justify the results obtained in this study. In this aspect, only a few studies explore similar applications and doses of biostimulants regarding the physiological aspects of plants, but there are reports of a greater tolerance in crops treated with biostimulants to stress conditions caused by water deficiency, using as an evaluation parameter, the production components and productivity of crops [38,39,40,41,42].

Except for the 2017/1 season, the same trend of behavior was observed through the doses of biostimulants in the seeds for the variables *gs* and *Ci*. This potentially indicates that the greater uptake of CO_2_ can be associated with the greater opening of the stomata and vice versa. Studies show that the sub-stomatal CO_2_ concentration in leaves generally increases with the increase in *gs* values, since the greater the stomatal opening, the greater the diffusion of CO_2_ to the substomatic chambers.

Considering the greater water availability during the first planting seasons in each year of experimental conduction (2016/1 and 2017/1), values for leaf area stood out in these periods, which were on average 16.5% and 34.5% higher than in 2016/2 and 2017/2, respectively (Figure 2G), where there is less water restriction due to the early sowing of second-crop corn in central Brazil [43], which favors the vegetative development of the crop and, consequently, the foliar growth of the plants.

In a study of corn seeds, Xue et al. [33] verified the importance of phytohormones such as auxin, gibberellin and cytokinin for germination and seedling development, which may explain the increase in the physiological yield of plants, stimulated by the biostimulant applied as seed treatment.

In addition to the 13.34% increase in the net photosynthesis rate, the presence of the biostimulant applied via foliar allowed the corn leaves to express their maximum photosynthetic potential when seeds were treated at lower doses. In this sense, it appears that plants that have an adequate hormonal balance have greater vegetative development [39], which triggers greater activity in physiological processes such as photosynthesis [44].

Assuming that stomatal conductance is related to the degree of stomata opening, the physiological behavior of the corn crop in this study corroborates the results obtained by Lima et al. [45], who observed that the opening and closing mechanisms of stomata can influence transpiration, and in cases of the greater physiological activity of plants, they are subject to loss of water to the environment, due to the opening of stomata to capture CO_2_. The act of maintaining the stomatal opening may also indicate that the treated plants have greater adaptability to environmental conditions and, consequently, to the sources of stress acting on them. Concomitantly, there is a significant reduction in reactive oxygen species and the production of antioxidant compounds that prevent cell degradation [4,5,42,44].

Given the increased demand for atmospheric CO_2_, and due to the greater hormonal stimulus with the application of biostimulants, the increase in photosynthetic activity causes the opening of stomatal clefts, reducing stomatal resistance to CO_2_ diffusion, making the loss of water through transpiration substantial under these conditions and enabling carbon fixation for the development of vegetative organs [16,46].

## 5. Conclusions

The application of biostimulants enhances the physiological performance of corn plants under both optimal environmental conditions and abiotic stress caused by water deficit.

Employing biostimulants in corn cultivation, whether applied to seeds or as a foliar spray, increases plant tolerance to water deficit, which typically occurs when crops are sown outside of the recommended timeframe.

Future studies may investigate the ideal doses and application timings of biostimulants for different edaphoclimatic conditions and explore a potential relationship between the application of biostimulants and the uptake of various nutrients by plants.

In general, the combined use of biostimulants on seeds and as a foliar treatment in corn cultivation boosts the physiological activity of plants by stimulating the photosynthetic process. Based on these data, the use of plant regulators can be a useful tool to mitigate the adverse effects of climate change on corn plants sown inside and outside the planting period.

## Figures and Tables

**Figure 1 plants-12-02569-f001:**
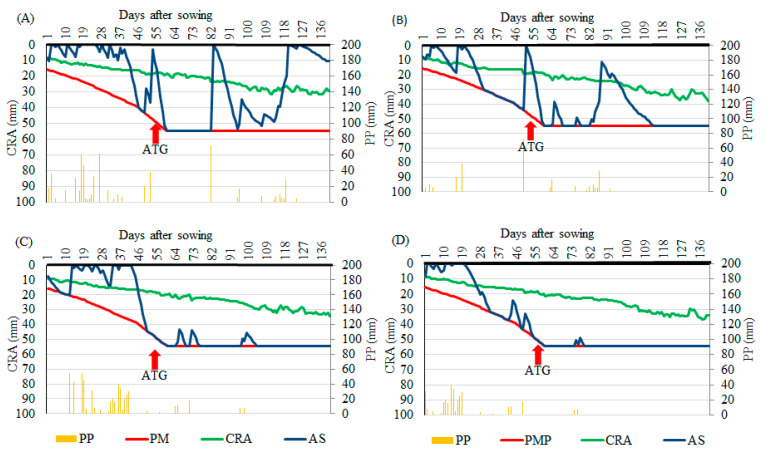
Rainfall (PP), permanent wilting point (PM), actual soil water capacity (CRA), soil water storage (AS) and evaluation of gas exchange in plants (ATG) for the cultivation of corn sown in February (**A**) and March (**B**) 2016 and February (**C**) and March (**D**) 2017 in Chapadão do Sul, MS.

**Figure 2 plants-12-02569-f002:**
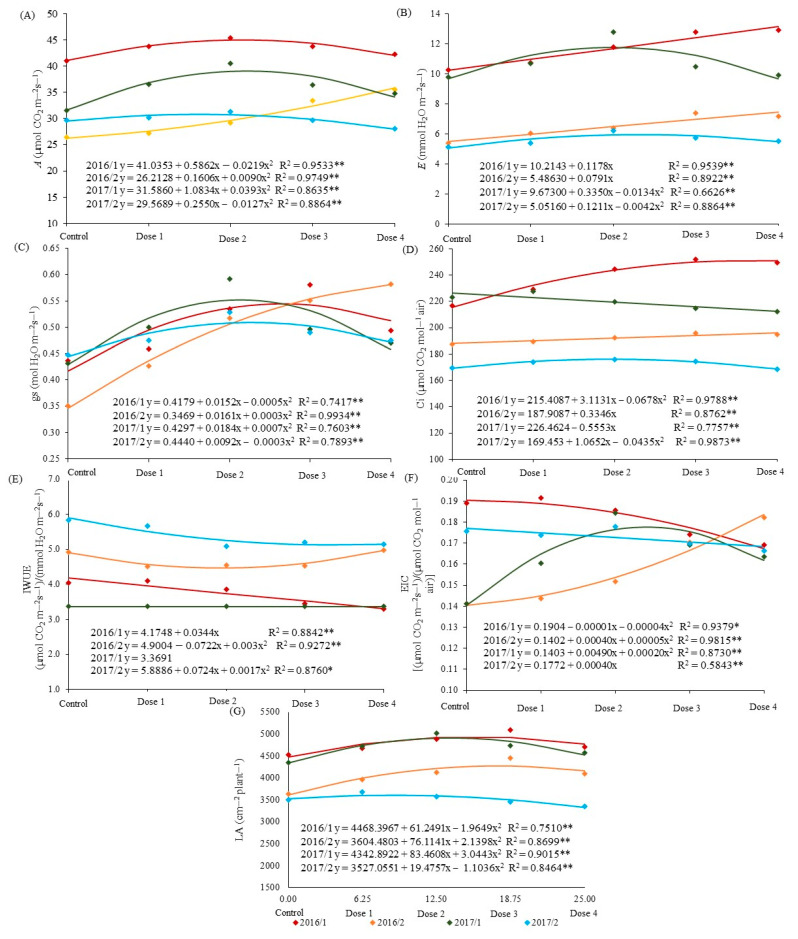
Regression curves for net photosynthesis rate (**A**), transpiration (**B**), stomatal conductance (**C**), sub-stomatal CO_2_ concentration (**D**), instantaneous water use efficiency (**E**), instantaneous carboxylation efficiency (**F**), and leaf area (**G**), as a function of the application of biostimulants in corn sown at different times. Chapadão do Sul, MS, 2016 and 2017. * or **—Significant at *p* ≤ 0.05 and ≤0.01 by F-test, respectively.

**Figure 3 plants-12-02569-f003:**
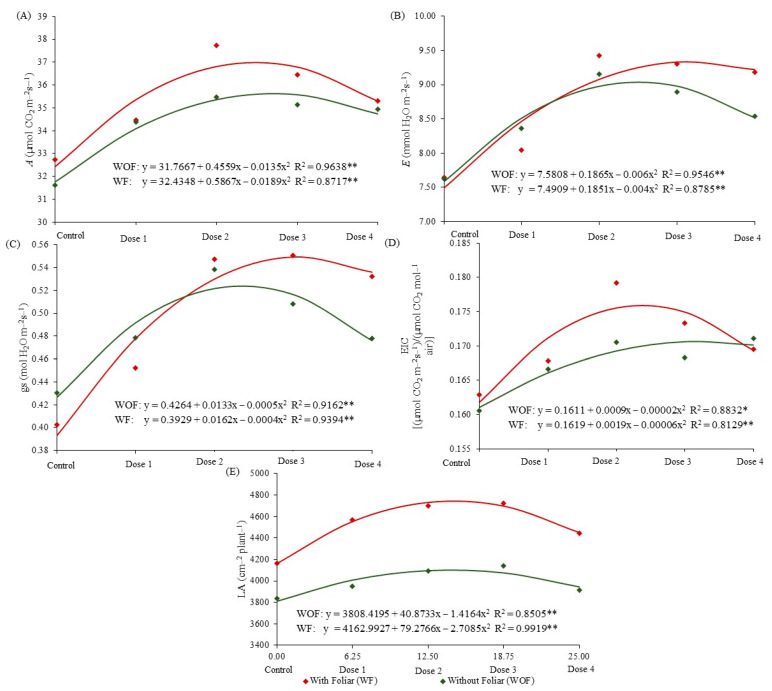
Regression curves for rate of net photosynthesis (**A**), transpiration (**B**), stomatal conductance (**C**), instantaneous carboxylation efficiency (EIC) (**D**), and leaf area (**E**), subjected to doses of the biostimulant on seeds with the presence or absence of foliar application of the same product: Chapadão do Sul, MS, 2016 and 2017. * or **—Significant at *p* ≤ 0.05 and ≤ 0.01 by F-test, respectively.

**Table 1 plants-12-02569-t001:** Analysis of variance and significance for net photosynthesis rate (*A*), transpiration (*E*), stomatal conductance (*gs*), sub-stomatal CO_2_ concentration (*Ci*), instantaneous water use efficiency (*IWUE*), instantaneous efficiency of carboxylation (*EIC*), and leaf area (*LA*), as a function of the application of biostimulants in corn sown at different times: Chapadão do Sul, MS, 2016 and 2017.

FV	LD	Mean Square						
		*A*	*E*	*gs*	*Ci*	*IWUE*	*EIC*	*LA*
Block	3	0.3	0.46	0.00006	71.33	0.17	0.00007	16,719.81
T	3	1565.78 **	367 **	0.003	33,927.14 **	33.39 **	0.004 **	13,817,860.87 **
F	1	42.96 **	1.65 *	0.004	197.1 **	0.04	0.0004 **	11,311,699.77 **
S	4	91.37 **	15.02 **	0.08 **	489.45 **	0.79 **	0.0008 **	980,519.65 **
T × F	3	72.31 **	0.8	0.01 **	99.24 **	1.59 **	0.002 **	1,411,731.74 **
T × S	12	50.51 **	4.91 **	0.02 **	602.65 **	0.60 **	0.002 **	228,291.09 **
F × S	4	5.91 **	1.10 *	0.01 **	18.11	0.18	0.0001 *	118,470.20 **
Error	129	1.37	0.39	0.002	20.81	0.12	0.00005	24,425.23
CV (%)		3.36	7.28	8.1	2.22	7.96	4.02	3.67
Average		34.82	8.62	0.49	205.65	4.3	0.17	4253.26

Sowing time (T), biostimulant foliar (F), and biostimulant in seeds (S). * and ** significant at 5 and 1% probability, respectively by the F test.

**Table 2 plants-12-02569-t002:** Mean values of net photosynthesis rate (*A*) (μmol of CO_2_ m^−2^ s^−1^), stomatal conductance (*gs*) (mol of H_2_O m^−2^ s^−1^), sub-stomatal CO_2_ concentration (*Ci*) (µmol CO_2_ mol^−1^ air), instantaneous water use efficiency (*IWUE*)—(μmol of CO_2_ m^−2^ s^−1^)/(mmol of H_2_O m^−2^ s^−1^), instantaneous carboxylation efficiency (*EIC*)—(μmol of CO_2_ m^−2^ s^−1^)/(μmol CO_2_ mol^−1^ air)—and leaf area (*LA*) (cm^2^ plant^−1^) as a function of the application of biostimulants in corn sown at different times in Chapadão do Sul, MS, 2016 and 2017.

Variable	Biostimulant	Sowing Time			
		2016/1	2016/2	2017/1	2017/2
*A*	With foliar	43.59 a A	32.74 a C	36.02 a B	29.01 b D
	Without foliar	42.89 a A	27.94 b D	35.82 a B	30.56 a C
*gs*	With foliar	0.51 a A	0.50 a A	0.52 a A	0.46 b B
	Without foliar	0.49 b A	0.47 a A	0.48 b A	0.51 a A
*Ci*	With foliar	239.11 a A	192.25 a C	222.96 a B	172.73 a D
	Without foliar	237.75 a A	191.93 a C	216.08 b B	172.42 a D
*IWUE*	With foliar	3.77 a B	4.87 a A	3.40 a C	5.09 b A
	Without foliar	3.72 a C	4.52 b B	3.34 a D	5.68 a A
*EIC*	With foliar	0.18 a A	0.17 a B	0.16 b C	0.17 b B
	Without foliar	0.18 a A	0.15 b C	0.17 a B	0.18 a A
*LA*	With foliar	4952.99 a A	4594.69 a B	4897.36 a A	3631.56 a C
	Without foliar	4593.98 b A	3514.09 b C	4449.28 b B	3392.11 b C

Means followed by equivalent capital letters in the row and lowercase in the column do not statistically differ from each other according to Tukey’s test at 5% probability.

## Data Availability

The datasets used and/or analyzed during the current study are available from the corresponding author on reasonable request.

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
