# Peer review of "Biostimulants in Corn Cultivation as a Means to Alleviate the Impacts of Irregular Water Regimes Induced by Climate Change"

_plants, 2023, doi:10.3390/plants12132569_

Round 1

Reviewer 1 Report

Review report (Round 1)

Journal: Plants (ISSN 2223-7747); Manuscript ID: plants-2440206

Type: Article

Title: Biostimulants in Corn Cultivation as a Means to Alleviate the Impacts of Irregular Water Regimes Induced by Climate Change

The authors reported experimental results of an application of “biostimulants” to maize. The results may have potential values. I agree that the topic is important. However, I have serious concerns (listed below), so that I think this manuscript is not suitable for publication in high-impact journals such as Plants (IF more than 4).

1. In Introduction, previous studies on application of “biostimulants” were not well reviewed in the Introduction. The most of the arguments appear in Introduction is about “general introduction” (climate change, corn, and plant stress physiology in general). Information about the background of present research is not well described in the Introduction.

The answers to the following questions should be explicitly clarified by citing relevant previous literatures.

-       Is the present results / hypothesis novel? If so, to what extent it is novel?

-       What are the relation between the present hypothesis and previous relevant studies?

Are there any studies on biostimulants? I think there are many studies.

I agree that some journals set their publication criterion as “technically sound”, meaning that novelty are not the issue. This is not the case for the present case of high-quality journals.

2. I found that the format of the manuscript does not follow the Author Guidelines. Some contents such as ”Author Contributions” are lacking. Unfortunately, therefore, in my opinion the manuscript is not suitable for publication in the present journal.

Author Response

In Introduction, previous studies on application of “biostimulants” were not well reviewed in the Introduction. The most of the arguments appear in Introduction is about “general introduction” (climate change, corn, and plant stress physiology in general). Information about the background of present research is not well described in the Introduction.

 We appreciate your comments and agree that the introduction of our study could benefit from a more in-depth analysis of the existing literature on the use of biostimulants as a possible strategy to address the challenges imposed by climate change. Therefore, we have reviewed the introduction section and incorporated more relevant references.

The answers to the following questions should be explicitly clarified by citing relevant previous literatures.

Is the present results / hypothesis novel? If so, to what extent it is novel?  What are the relation between the present hypothesis and previous relevant studies?

We would like to state that our study was motivated by a practical problem that has been observed in the field, both in Brazil and in the United States. The planting dates recommended by governments have not been aligning with the ideal climatic conditions. This suggests that climate change is altering the expected seasonal pattern of rainfall, consequently making it difficult for farmers to make decisions. Therefore, our hypothesis and research represent an innovative contribution to this field of study, as our work specifically focuses on evaluating how biostimulants can help mitigate this problem in corn cultivation.

Are there any studies on biostimulants? I think there are many studies.

we have reviewed the introduction section and incorporated more relevant references. 

  1. I found that the format of the manuscript does not follow the Author Guidelines. Some contents such as ”Author Contributions” are lacking. Unfortunately, therefore, in my opinion the manuscript is not suitable for publication in the present journal.

We thank you for pointing out the lack of the "Author Contributions" section.

We will be careful to review and correct the manuscript format, so that it is fully aligned with the guidelines of the journal.

Reviewer 2 Report

Dear Authors, I revised the manuscript, "Biostimulants in Corn Cultivation as a Means to Alleviate the Impacts of Irregular Water Regimes Induced by Climate Change" and in my opinion, only minor revisions are required in order to process the paper for publication. First of all, you should change the reference style in MDPI form. Lines 69–78: Please improve this part of the text and state in a more clear way what the aims of this study were. In the materials and methods section, there is a typographic error in Line 103: 'plant. 0.2652'. 

Quality of English Language was fine.

Author Response

Dear Reviewer,

We sincerely appreciate the time and effort you took to review our manuscript. Your suggestions and comments helped us to improve the quality of our work.

In response to your observations, we would like to inform you that we have made the necessary changes as requested:

We updated the reference style to the MDPI format. We appreciate you pointing out this issue.

We have improved the section between lines 69-78 to clarify the aims of this study. We hope it is now clearer and more precise.

We corrected the typographical error on line 103, "plant. 0.2652". Thank you for identifying this error.

We thank you again for your valuable contribution to enhancing our manuscript. We are confident that these changes will make our work stronger and more suitable for publication.

Best regards,
Renato

Reviewer 3 Report

The work is interesting and well-presented. However, I think that it will be improved: I suggest some little notes that should be useful for improving it.

In the abstract lines 37-38-39, you repeat the aim of the trial that you define above. In my opinion, maybe more interesting that you explain the differences found in the trial both for seed and foliar application of biostimulant, that occur in more restrictive conditions to vegetative development in relation to rainfall and water deficit... i.e. in leaf area or in gas exchange results that showed better performances in stressful conditions for plants. Add shortly something of your results, also reporting aspects related to stimulation of the photosynthetic process!

Some little corrections or suggestions:

Line 76. You repeat conditions... that is the last term used in the previous sentence. Probably you can substitute with ... These situations..

Line 105:  you report M.O. without specifying what it means (May be the organic matter? In this case it should be inverted in O.M.)! Please define it. It is reported also in line 108.

Line 122: Please correct m2 with m2

Line 150: .infrared... I think that the sentence should be correct... by using infrared...

Lime 154: there is a space more... m -2 - Please correct with m-2

Line 162: I suggest use choosing instead using

Line 183: For a better comprehension I think that is better to add at the end of the sentence ...area (Table 1).

Line 199: .also .. I think that the sentence should be correct...

Line 202, 266, and 269: CO2 Please change with CO2

Line 206: In my opinion, the title of the table should be changed in this way: "Mean values of.... (report the same parameters...) in relation to

LIne 272, and  359 (Figure 2, and Figure 3)In my opinion, the title should be changed by adding some information regarding the curve reported.  I think you should explain that are referred to regression analysis. Graphically, the figures may be improved without the values of doses 0.00 and 25.00 at the extreme of the x-axis. You can treat doses as  "text" separating them in a better way.

Line 323: The sentence reported is too long. Please rewrite it, dividing it into more sentences.

Line 336, and 338: H2O Please change with H2O

Discussion: You report some nutritional implications due to biostimulants application in other works. In your work, no data are reported about it. So, in order to give a high scientific value to your paper, you can propose future studies where will be scheduled these assessments. The same happens with partitioning (you speak about root development, a parameter that you did not measure..)

Line 461: save money. the water available in the soil... It's not easy to understand if you don't link the two sentences... 

Line 478: plantingseasons change with planting seasons

Line 514 : substantial. under  Please cut the point!

Author Response

Dear Reviewer,

Thank you very much for your detailed review of our manuscript. We are grateful for your constructive comments and suggestions, all of which have been considered and implemented to improve our work.

Regarding your specific comments:

In the abstract lines 37-38-39, you repeat the aim of the trial that you define above. In my opinion, maybe more interesting that you explain the differences found in the trial both for seed and foliar application of biostimulant, that occur in more restrictive conditions to vegetative development in relation to rainfall and water deficit... i.e. in leaf area or in gas exchange results that showed better performances in stressful conditions for plants. Add shortly something of your results, also reporting aspects related to stimulation of the photosynthetic process! Some little corrections or suggestions:

In the introduction and summary, we modified the text in an attempt to meet your request and that of the other reviewers to provide more clarity to the study.

About lines: We have made the necessary corrections as per your suggestions on the other lines pointed out. Each line has been reviewed and the errors have been corrected. 

Discussion: You report some nutritional implications due to biostimulants application in other works. In your work, no data are reported about it. So, in order to give a high scientific value to your paper, you can propose future studies where will be scheduled these assessments. The same happens with partitioning (you speak about root development, a parameter that you did not measure..)

In the discussion and conclusion section, we included some statements about future studies that will focus on the nutritional implications of the application of biostimulants, as well as a discussion about root development and plant morphology.

We sincerely appreciate your valuable contribution and believe that these modifications have significantly improved the quality and readability of our manuscript. We hope that our revisions meet with your approval.

Best regards,
Renato

Round 2

Reviewer 1 Report

1. In my previous comments, I have asked the following three questions:

Is the present results / hypothesis novel? If so, to what extent it is novel?  What are the relation between the present hypothesis and previous relevant studies?

Authors’ reply is as follows:

We would like to state that our study was motivated by a practical problem that has been observed in the field, both in Brazil and in the United States. The planting dates recommended by governments have not been aligning with the ideal climatic conditions. This suggests that climate change is altering the expected seasonal pattern of rainfall, consequently making it difficult for farmers to make decisions. Therefore, our hypothesis and research represent an innovative contribution to this field of study, as our work specifically focuses on evaluating how biostimulants can help mitigate this problem in corn cultivation.”

(My comments) Unfortunately, I think the authors do not answer to my questions. I did not ask the importance of the present topic. I asked about novelty. Because this is second-round review, I am very sorry to concluded that I cannot recommend publication of this manuscript.

2. Did the authors use “tack-change” function of MS-WORD? I cannot see the changed part.

Author Response

In my previous comments, I have asked the following three questions:

 Is the present results / hypothesis novel? If so, to what extent it is novel?  What are the relation between the present hypothesis and previous relevant studies?

 Authors’ reply is as follows:

 We would like to state that our study was motivated by a practical problem that has been observed in the field, both in Brazil and in the United States. The planting dates recommended by governments have not been aligning with the ideal climatic conditions. This suggests that climate change is altering the expected seasonal pattern of rainfall, consequently making it difficult for farmers to make decisions. Therefore, our hypothesis and research represent an innovative contribution to this field of study, as our work specifically focuses on evaluating how biostimulants can help mitigate this problem in corn cultivation.”

 (My comments) Unfortunately, I think the authors do not answer to my questions. I did not ask the importance of the present topic. I asked about novelty. Because this is second-round review, I am very sorry to concluded that I cannot recommend publication of this manuscript.

Response: Thanks for valuable comment. Yes, we think that our study is novel and bring important information on the role of biostimulants (plant growth regulators) in mitigate effect of climate changes on corn cultivation under filed conditions.

Both lab and field studies are important in advancing our understanding of plant growth regulation. Although there are many lab studies on the stress mitigating effect of plant growth regulators (Abdel Megeed ety al., 2021, Zulfiqar et al., 2020), this effect is hardly study under field conditions.

Compared to lab studies, field studies provide a more holistic perspective, considering the complexities and variability of real-world conditions. Thus, field studies are essential for understanding the complex interactions between growth regulator hormones and plant stress responses. In this regard, our study also focuses on the evaluation and optimization of application methodologies, providing valuable insights into the practical applications of a specific proportion mixture of growth-promoting hormones (kinetin, gibberellin, indolebutyric acid)  for stress tolerance in corn agriculture, taking into account the genetic, environmental, and ecological complexities that exist in natural systems.

Details about the novelty of the work have been added.

Zulfiqar, F., Casadesús, A., Brockman, H., & Munné-Bosch, S. (2020). An overview of plant-based natural biostimulants for sustainable horticulture with a particular focus on moringa leaf extracts. Plant Science, 295, 110194.

Abdel Megeed, T. M., H. S. Gharib, E. M. Hafez, and A. El-Sayed. "Effect of some plant growth regulators and biostimulants on the productivity of Sakha108 rice plant (Oryza sativa L.) under different water stress conditions." Appl. Ecol. Environ. Res 19 (2021): 2859-2878.

Malik, A., Mor, V. S., Tokas, J., Punia, H., Malik, S., Malik, K., Sangwan, S., Tomar, S., Singh, P., Singh, N., Himangini, Vikram, Nidhi, Singh, G., Vikram, Kumar, V., Sandhya, & Karwasra, A. (2020). Biostimulant-Treated Seedlings under Sustainable Agriculture: A Global Perspective Facing Climate Change. Agronomy 2021, Vol. 11, Page 14, 11(1), 14. https://doi.org/10.3390/AGRONOMY11010014

Baum, M. E., Licht, M. A., Huber, I., & Archontoulis, S. v. (2020). Impacts of climate change on the optimum planting date of different maize cultivars in the central US Corn Belt. European Journal of Agronomy, 119, 126101. https://doi.org/10.1016/J.EJA.2020.126101

  1. Did the authors use “tack-change” function of MS-WORD? I cannot see the changed part.

Response: Thanks for comment, in the revised version we highlighted all changes to easly track them.

Round 3

Reviewer 1 Report

I have already recommended "reject" for this manuscript for two times. I think it is unusual for high-impact journal to give a round review, when the authors did not responsible properly to my first-round comments in the second-round review. I think no more review round is needed, and that it is the Editor's responsibility to decide whether the manuscript should be published.

Author Response

Dear reviewer,

We greatly appreciate your time and the effort put into reviewing our manuscript. We recognize the importance of your notes and agree that they were crucial to the improvement of our work.

We regret any misunderstanding or dissatisfaction caused by our previous responses. However, we would like to assure you that we take all of your suggestions seriously and that they were carefully considered in the revision of our manuscript.

We
believe that the manuscript has been significantly improved, through the notes from all reviewers, including yours.

We understand and respect your position and agree that it is now up to the editor to make the final decision on the publication of the manuscript.

We thank you again for your time and your valuable contribution to our work. Sincerely,
Renato
